# Non-Projective Two-Weight Codes

**DOI:** 10.3390/e26040289

**Published:** 2024-03-27

**Authors:** Sascha Kurz

**Affiliations:** 1Mathematisches Institut, Universität Bayreuth, D-95440 Bayreuth, Germany; sascha.kurz@uni-bayreuth.de; 2Department of Data Science, Friedrich-Alexander-Universitäät Erlangen-Nürnberg, D-91058 Erlangen, Germany; sascha.kurz@fau.de

**Keywords:** linear codes, two-weight codes, two-character sets, 94B05, 05B25, 68R01

## Abstract

It has been known since the 1970’s that the difference of the non-zero weights of a projective $F_{q}$-linear two-weight code has to be a power of the characteristic of the underlying field. Here, we study non-projective two-weight codes and, e.g., show the same result under mild extra conditions. For small dimensions we give exhaustive enumerations of the feasible parameters in the binary case.

## 1. Introduction

It has been known since the 1970s that the two non-zero weights of a projective $F_{q}$-linear two-weight code *C* can be written as $w_{1}=up^{t}$ and $w_{2}=\left(u+1\right)p^{t}$, where $u\in N_{\geq 1}$ and *p* is the characteristic of the underlying finite field $F_{q}$; see Corollary 2 [1]. So, especially the weight difference $w_{2}-w_{1}$ is a power of the characteristic *p*. Here, we want to consider $F_{q}$-linear two-weight codes *C* with non-zero weights $w_{1}<w_{2}$ which are not necessarily projective. In [2], it was observed that if $w_{2}-w_{1}$ is not a power of the characteristic *p*, then the code *C* has to be non-projective, which settles a question in [3]. Here, we prove the stronger statement that *C* is repetitive, i.e., *C* is the *l*-fold repetition of a smaller two-weight code $C^{\prime }$, where *l* is the largest factor of $w_{2}-w_{1}$ that is coprime to the field size *q*, if *C* does not have full support, cf. [4]. Moreover, if a two-weight code *C* is non-repetitive and does not have full support, then its two non-zero weights can be written as $w_{1}=up^{t}$ and $w_{2}=\left(u+1\right)p^{t}$, where again *p* is the characteristic of the underlying finite field $F_{q}$; see Theorem 3.

Constructions for projective two-weight codes can be found in the classical survey paper [5]. Many research papers considered these objects since they, e.g., yield strongly regular graphs (srgs), and we refer the reader to [6] for a corresponding monograph on srgs. For a few more recent papers on constructions for projective two-weight codes, we refer, e.g., to [7,8,9,10]. In, e.g., [9], the author uses geometric language and speaks of constructions for two-character sets, i.e., sets of points in a projective space PG(k−1,q) with just two different hyperplane multiplicities; call them *s* and *t*. In general, each (full-length) linear code is in one-to-one correspondence to a (spanning) multiset of points in some projective space PG(k−1,q). Here, we will also mainly use the geometric language and consider a few general constructions for two-character multisets of points corresponding to two-weight codes (possibly non-projective). For each subset $\bar{H}$ of hyperplanes in PG(k−1,q) we construct a multiset of points $M(\bar{H})$ such that all hyperplanes $H\in \bar{H}$ have the same multiplicity, say *s*, and also all other hyperplanes $H\notin \bar{H}$ have the same multiplicity, say *t*. Actually, we characterize the full set of such multisets with at most two different hyperplane multiplicities given $\bar{H}$; see Theorems 1 and 2. Using this correspondence we have classified all two-weight codes up to symmetry for small parameters. For projective two-weight codes such enumerations can be found in [11].

Brouwer and van Eupen give a correspondence between arbitrary projective codes and arbitrary two-weight codes via the so-called BvE dual transform. The correspondence can be said to be 1–1, even though there are choices for the involved parameters to be made in both directions. In [12], the dual transform was, e.g., applied to the unique projective ${\left[16,5,9\right]}_{3}$-code. For parameters $\alpha =\frac{1}{3}$ and β=−3 the result is a ${\left[69,5,45\right]}_{3}$ two-weight code, and for $\alpha =-\frac{1}{3}$ and β=5 the result is a ${\left[173,5,108\right]}_{3}$ two-weight code. This resembles the fact that we have some freedom when constructing a two-weight code from a given projective code, e.g., we can take complements or add simplex codes of the same dimension. Our obtained results may be rephrased in the language of the BvE dual transform by restricting to a canonical choice of the involved parameters. For further literature on the dual transform, see, e.g., [12,13,14,15]. For a variant that is rather close to our presentation we refer the reader to [16].

With respect to further related studies in the literature we remark that a special subclass of (non-projective) two-weight codes was completely characterized in [17]. A conjecture by Vega [18] states that all two-weight cyclic codes are the “known” ones, cf. [19]. Another stream of the literature considers the problem of whether all projective two-weight codes that have the parameters of partial *k*-spreads indeed have to be partial *k*-spreads. Those results can be found in papers considering extendability results for partial *k*-spreads or classifying minihypers; see, e.g., [20]. Two-weight codes have also been considered over rings instead of finite fields; see, e.g., [21].

The remaining part of this paper is structured as follows. In Section 2, we introduce the necessary preliminaries for linear two-weight codes and their geometric counterpart called two-character multisets in projective spaces. In general, multisets of points, corresponding to general linear codes, can be described via so-called characteristic functions and we collect some of their properties in Section 3. Examples and constructions for two-character multisets are given in Section 4. In Section 5, we present our main results. We close with enumeration of the results for two-character multisets in PG(k−1,q) for small parameters in Section 6. We will mainly use geometric language and arguments. For the ease of the reader we only use elementary arguments and give (almost) all details.

## 2. Preliminaries

An ${\left[n,k\right]}_{q}$-code *C* is a *k*-dimensional subspace of $F_{q}^{n}$, i.e., *C* is assumed to be $F_{q}$-linear. Here, *n* is called the length and *k* is called the dimension of *C*. Elements c∈C are called codewords and the weight wt(c) of a codeword is given by the number of non-zero coordinates. Clearly, the all-zero vector 0 has weight zero and all other codewords have a positive integer weight. A two-weight code is a linear code with exactly two non-zero weights. A generator matrix for *C* is a k×n matrix *G* such that its rows span *C*. We say that *C* is of full length if for each index 1≤i≤n there exists a codeword c∈C whose *i*th coordinate $c_{i}$ is non-zero, i.e., all columns of a generator matrix of *C* are non-zero. The dual code $C^{⊥}$ of *C* is the (n−k)-dimensional code consisting of the vectors orthogonal to all codewords of *C* with respect to the inner product $\langle u,v\rangle =\sum_{i=1}^{n}u_{i}v_{i}$.

Now, let *C* be a full-length ${\left[n,k\right]}_{q}$-code with generator matrix *G*. Each column *g* of *G* is an element of $F_{q}^{k}$ and since g≠0 we can consider 〈g〉 as a point in the projective space PG(k−1,q). Using the geometric language we call 1-, 2-, 3-, and (k−1)-dimensional subspaces of $F_{q}^{k}$ points, lines, planes, and hyperplanes in PG(k−1,q). Instead of an *l*-dimensional space we also speak of an *l*-space. By P we denote the set of points and by H we denote the set of hyperplanes of PG(k−1,q) whenever *k* and *q* are clear from the context. A multiset of points in PG(k−1,q) is a mapping M:P→N, i.e., to each point P∈P we assign its multiplicity M(P)∈N. By $\#M=\sum_{P\in P}M\left(P\right)$ we denote the cardinality of M. The support supp(M) is the set of all points with non-zero multiplicity. We say that M is spanning if the set of points in the support of M span PG(k−1,q). Clearly permuting columns of a generator matrix *G* or multiplying some columns with non-zero elements in $F_{q}^{★}:=F_{q}\backslash \left\{0\right\}$ yields an equivalent code. In addition to that we obtain a one-to-one correspondence between full-length ${\left[n,k\right]}_{q}$-codes and spanning multisets of points M in PG(k−1,q) with cardinality #M=n. Moreover, two linear ${\left[n,k\right]}_{q}$-codes *C* and $C^{\prime }$ are equivalent if their corresponding multisets of points M and $M^{\prime }$ are. For details we refer, e.g., to [22]. A linear code *C* is projective if its corresponding multiset of points satisfies M(P)∈{0,1} for all P∈P. We also speak of a set of points in this case. The multisets of points with M(P)=0 for all P∈P are called trivial.

Geometrically, for a non-zero codeword c∈C the set $c\cdot F_{q}^{★}$ corresponds to a hyperplane H∈H and wt(c)=#M−M(H), where we extend the function M additively, i.e., $M\left(S\right):=\sum_{P\in S}M\left(P\right)$ for every subset S⊆P of points. We call M(H) the multiplicity of hyperplane H∈H and have M(V)=#M for the entire ambient space V:=P. The number of hyperplanes #H, as well as the number of points #P, in PG(k−1,q) is given by ${\left[k\right]}_{q}:=\frac{q^{k}-1}{q-1}$. A two-character multiset is a multiset of points M such that exactly two different hyperplane multiplicities M(H) occur. i.e., a multiset of points M is a two-character multiset if its corresponding code *C* is a two-weight code. If M actually is a set of points, i.e., if we have M(P)∈{0,1} for all points P∈P, then we speak of a two-character set. We say that an ${\left[n,k\right]}_{q}$-code *C* is Δ-divisible if the weights of all codewords are divisible by Δ. A multiset of points M is called Δ-divisible if the corresponding linear code is. More directly, a multiset of points M is Δ-divisible if we have M(H)≡#M(modΔ) for all H∈H.

A one-weight code is an ${\left[n,k\right]}_{q}$-code *C* such that all non-zero codewords have the same weight *w*. One-weight codes have been completely classified in [23] and are given by repetitions of so-called simplex codes. Geometrically, the multiset of points M in PG(k−1,q) corresponding to a one-weight code *C* satisfies M(P)=l for all P∈P, i.e., we have $\#M=n={\left[k\right]}_{q}\cdot l$, $M\left(H\right)={\left[k-1\right]}_{q}\cdot l$ for all H∈H, and $w=\#M-M\left(H\right)=q^{k-1}\cdot l$. We say that a linear ${\left[n,k\right]}_{q}$-code *C* is repetitive if it is the *l*-fold repetition of an ${\left[n/l,k\right]}_{q}$-code $C^{\prime }$, where l>1, and non-repetitive otherwise. A given multiset of points M is called repeated if its corresponding code is. More directly, a non-trivial multiset of points M is repeated if the greatest common divisor of all point multiplicities is larger than one. We say that a multiset of points M or its corresponding linear code *C* has full support if supp(M)=P, i.e., if M(P)>0 for all P∈P. So, for each non-repetitive one-weight code *C* with length *n*, dimension *k*, and non-zero weight *w* we have $n={\left[k\right]}_{q}$ and $w=q^{k-1}$. Each non-trivial one-weight code, i.e., one with dimension at least 1, has full support. The aim of this paper is to characterize the possible parameters of non-repetitive two-weight codes (with or without full support). For the correspondence between ${\left[n,k\right]}_{q}$-codes and multisets of points M in PG(k−1,q) we have assumed that M is spanning. If M is not spanning, then there exists a hyperplane containing the entire support, so that M is two-character multiset if M induces a one-character multiset in the span of supp(M), cf. Proposition 1. The structure of the set of all two-character multisets where the larger hyperplane multiplicity is attained for a prescribed subset of the hyperplanes is considered in Section 5.

## 3. Characteristic Functions

Fixing the field size *q* and the dimension *k* of the ambient space, a multiset of points in PG(k−1,q) is a mapping M:P→N. By F we denote the Q-vector space consisting of all functions F:P→Q, where addition and scalar multiplication is defined pointwise. i.e., $\left(F_{1}+F_{2}\right)\left(P\right):=F_{1}\left(P\right)+F_{2}\left(P\right)$ and $\left(x\cdot F_{1}\right)\left(P\right):=x\cdot F_{1}\left(P\right)$ for all P∈P, where $F_{1},F_{2}\in F$, and x∈Q are arbitrary. For each non-empty subset S⊆P the characteristic function $\chi_{S}$ is defined by $\chi_{S}\left(P\right)=1$ if P∈S and $\chi_{S}\left(P\right)$ otherwise. Clearly the set of functions $\chi_{P}$ for all P∈P forms a basis of F for ambient space PG(k−1,q) for all k≥1. Note that there are no hyperplanes if k=1 and hyperplanes coincide with points for k=2. We also extend the functions F∈F additively, i.e., we set $F\left(S\right)=\sum_{P\in S}F\left(P\right)$ for all S⊆P. Our next aim is to show the well-known fact that also the set of functions $\chi_{H}$ for all hyperplanes H∈H forms a basis of F for ambient space PG(k−1,q) for all k≥2. In other words, also M(P) can be reconstructed from the M(H):

**Lemma 1.** 
*Let M∈F for ambient space PG(k−1,q), where k≥2. Then, we have*

(1)
$$M\left(P\right)=\sum_{H\in H\;:\;P\in H}\frac{1}{{\left[k-1\right]}_{q}}\cdot M\left(H\right)+\sum_{H\in H\;:\;P\notin H}\frac{1}{q^{k-1}}\cdot \left(\frac{1}{{\left[k-1\right]}_{q}}-1\right)\cdot M\left(H\right)$$

*for all points P∈P.*


**Proof.** Without loss of generality we assume k≥3. Since each point $P^{\prime }\in P$ is contained in ${\left[k-1\right]}_{q}$ of the $\#H={\left[k\right]}_{q}$ hyperplanes and each point $P^{\prime }\neq P$ is contained in ${\left[k-2\right]}_{q}$ of the ${\left[k-1\right]}_{q}$ hyperplanes that contain *P*, we have
$$\sum_{H\in H\;:\;P\in H}M\left(H\right)={\left[k-2\right]}_{q}\cdot \left|M\right|+\left({\left[k-1\right]}_{q}-{\left[k-2\right]}_{q}\right)M\left(P\right)={\left[k-2\right]}_{q}\cdot \left|M\right|+q^{k-2}M\left(P\right)$$
so that
$$\sum_{H\in H\;:\;P\in H}M\left(H\right)-\frac{{\left[k-2\right]}_{q}}{{\left[k-1\right]}_{q}}\cdot \sum_{H\in H}M\left(H\right)=q^{k-2}M\left(P\right)$$
using ${\left[k-1\right]}_{q}\cdot \#M=\sum_{H\in H}M\left(H\right)$. Thus, we can conclude the stated formula using
$$\frac{1}{q^{k-2}}\cdot \left(1-\frac{{\left[k-2\right]}_{q}}{{\left[k-1\right]}_{q}}\right)=\frac{1}{q^{k-2}}\cdot \frac{{\left[k-1\right]}_{q}-{\left[k-2\right]}_{q}}{{\left[k-1\right]}_{q}}=\frac{1}{{\left[k-1\right]}_{q}}$$
and
$$-\frac{{\left[k-2\right]}_{q}}{{\left[k-1\right]}_{q}\cdot q^{k-2}}=\frac{1-{\left[k-1\right]}_{q}}{{\left[k-1\right]}_{q}\cdot q^{k-1}}=\frac{1}{q^{k-1}}\cdot \left(\frac{1}{{\left[k-1\right]}_{q}}-1\right).$$□

As an example we state that in PG(3−1,2) we have
$$M\left(P\right)=\frac{1}{3}\cdot \sum_{H\in H\;:\;P\in H}M\left(H\right)-\frac{1}{6}\cdot \sum_{H\in H\;:\;P\notin H}M\left(H\right).$$

**Lemma 2.** 
*Let M∈F for ambient space PG(k−1,q), where k≥2. Then, there exist $\alpha_{H}\in Q$ for all hyperplanes H∈H such that*

(2)
$$M=\sum_{H\in H}\alpha_{H}\cdot \chi_{H}.$$

*Moreover, the coefficients $\alpha_{H}$ are uniquely determined by M.*


**Proof.** Each point P∈P is contained in ${\left[k-1\right]}_{q}$ hyperplanes and for each point Q≠P there are exactly ${\left[k-2\right]}_{q}$ hyperplanes that contain both *P* and *Q*, so that
$$\sum_{H\in H\;:\;P\in H}\chi_{H}-\frac{{\left[k-2\right]}_{q}}{{\left[k-1\right]}_{q}}\cdot \sum_{H\in H}\chi_{H}=q^{k-2}\cdot \chi_{P}.$$
Using
$$M=\sum_{P\in P}M\left(P\right)\cdot \chi_{P}$$
we conclude the existence of the $\alpha_{H}\in Q$. Since the functions ${\left(\chi_{P}\right)}_{P\in P}$ form a basis of the Q-vector space F, which is also generated by the functions ${\left(\chi_{H}\right)}_{H\in H}$, counting $\#P={\left[k\right]}_{q}=\#H$ yields that also ${\left(\chi_{H}\right)}_{H\in H}$ forms a basis and the coefficients $\alpha_{H}$ are uniquely determined by M. □

If M∈F is given by the representation
$$M=\sum_{P\in P}\alpha_{P}\cdot \chi_{P}$$
with $\alpha_{P}\in Q$ we can easily decide whether M is a multiset of points. The necessary and sufficient conditions are given by $\alpha_{P}\in N$ for all P∈P (including the case of a trivial multiset of points). If a multiset of points is characterized by coefficients $\alpha_{H}$ for all hyperplanes H∈H, as in Lemma 2, then some $\alpha_{H}$ may be fractional or negative. For two-character multisets we will construct a different unique representation, involving the characteristic functions $\chi_{H}$ of hyperplanes; see Theorem 1.

Let us state a few observations about operations for multisets of points that yield multisets of points again.

**Lemma 3.** 
*For two multisets of points $M_{1}$ and $M_{2}$ of PG(k−1,q) and each non-negative integer n∈N the functions $M_{1}+M_{2}$ and $n\cdot M_{1}$ are multisets of points of PG(k−1,q).*


In order to say something about the subtraction of multisets of points we denote the minimum point multiplicity of a multiset of points M by μ(M) and the maximum point multiplicity by γ(M). Whenever M is clear from the context we also just write μ and γ instead of μ(M) and μ(γ).

**Lemma 4.** 
*Let $M_{1}$ and $M_{2}$ be two multisets of points of PG(k−1,q). If $\mu \left(M_{1}\right)\geq \gamma \left(M_{2}\right)$, then $M_{1}-M_{2}$ is a multiset of points of PG(k−1,q).*


**Definition 1.** 
*Let M be a multiset of points in PG(k−1,q). If l is an integer with l≥γ(M), then the l-complement $M^{l-C}$ of M is defined by $M^{l-C}\left(P\right):=l-M\left(P\right)$ for all points P∈P.*


One can easily check that $M^{l-C}$ is a multiset of points with cardinality $l\cdot {\left[k\right]}_{q}-\#M$, maximum point multiplicity $\gamma \;\left(M^{l-C}\right)=l-\mu \left(M\right)$, and minimum point multiplicity $\mu \;\left(M^{l-C}\right)=l-\gamma \left(M\right)$. Using characteristic functions we can write $M^{l-C}=l\cdot \chi_{V}-M$, where V=P denotes the set of all points of the ambient space.

Given an arbitrary function M∈F there always exist α∈Q\{0} and β∈Z such that $\alpha \cdot M+\beta \cdot \chi_{V}$ is a multiset of points.

## 4. Examples and Constructions for Two-Character Multisets

The aim of this section is to list a few easy constructions for two-character multisets of points M in PG(k−1,q). We will always abbreviate n=#M and denote the two occurring hyperplane multiplicities by *s* and *t*, where we assume s>t by convention.

**Proposition 1.** 
*For integers 1≤l<k, let L be an arbitrary l-space in PG(k−1,q). Then, $\chi_{L}$ is a two-character set with $n={\left[l\right]}_{q}$, γ=1, μ=0, $s={\left[l\right]}_{q}$, and $t={\left[l-1\right]}_{q}$.*


Note that for the case l=k we have the one-character set $\chi_{V}$, which can be combined with any two-character multiset.

**Lemma 5.** 
*Let M be a two-character multiset of points in PG(k−1,q). Then, for each integer 0≤a≤μ(M), each b∈N, and each integer c≥γ(M) the functions $M-a\cdot \chi_{V}$, $M+b\cdot \chi_{V}$, b·M, and $c\cdot \chi_{V}-M$ are two-character multisets of points.*


For the first and the fourth construction we also spell out the implications for the parameters of a given two-character multiset:

**Lemma 6.** 
*Let M be a multiset of points in PG(k−1,q) such that M(H)∈{s,t} for every hyperplane H∈H. If M(P)≥l for every point P∈P, i.e., l≤μ(M), then $M^{\prime }:=M-l\cdot \chi_{V}$ is a multiset of points in PG(k−1,q) such that $M^{\prime }\left(H\right)\in \left\{s-{\left[k-1\right]}_{q}\cdot l,t-{\left[k-1\right]}_{q}\cdot l\right\}$ for every hyperplane H∈H.*


**Lemma 7.** 
*Let M be a multiset of points in PG(k−1,q) such that M(H)∈{s,t} for every hyperplane H∈H. If M(P)≤u, i.e., ≤γ(M) for every point P∈P, then the u-complement $M^{\prime }:=u\cdot \chi_{V}-M$ of M is a multiset of points in PG(k−1,q) such that $M^{\prime }\left(H\right)\in \left\{u[k-1]-s,u[k-1]-t\right\}$ for every hyperplane H∈H.*


We can also use two (almost) arbitrary subspaces to construct two-character multisets:

**Proposition 2.** 
*Let a≥b≥1 and 0≤i≤b−1 be arbitrary integers, A be an a-space, and B be a b-space with dim(A∩B)=i in PG(k−1,q), where k=a+b−i. Then, $M=\chi_{A}+q^{a-b}\cdot \chi_{B}$ satisfies M(H)∈{s,t} for all H∈H, where $s={\left[a-1\right]}_{q}+q^{a-b}\cdot {\left[b-1\right]}_{q}$ and $t=s+q^{a-1}$. If i=0, then $\gamma =q^{a-b}$, and $\gamma =q^{a-b}+1$ otherwise. In general, we have $n={\left[a\right]}_{q}+q^{a-b}\cdot {\left[b\right]}_{q}$ and μ=0.*


**Proof.** For each H∈H we have $M\left(H\cap A\right)\in \left\{{\left[a-1\right]}_{q},{\left[a\right]}_{q}\right\}$ and $M\left(H\cap B\right)\in \left\{{\left[b-1\right]}_{q},{\left[b\right]}_{q}\right\}$. Noting that we cannot have both $M\left(H\cap A\right)={\left[a\right]}_{q}$ and $M\left(H\cap B\right)={\left[b\right]}_{q}$, we conclude $M\left(H\right)\in \left\{{\left[a-1\right]}_{q}+q^{a-b}\cdot {\left[b-1\right]}_{q},{\left[a\right]}_{q}+q^{a-b}\cdot {\left[b-1\right]}_{q},{\left[a-1\right]}_{q}+q^{a-b}\cdot {\left[b\right]}_{q}\right\}=\left\{s,t\right\}$. □

A partial *k*-spread is a set of *k*-spaces in PG(v−1,q) with pairwise trivial intersection.

**Proposition 3.** 
*Let $S_{1},...,S_{r}$ be a partial parallelism of PG(2k−1,q), i.e., the $S_{i}$ are partial k-spreads that are pairwise disjoint. Then,*

$$M=\sum_{i=1}^{r}\sum_{S\in S_{i}}\chi_{S}$$

*is a two-character multiset of PG(2k−1,q) with $n=r\cdot {\left[k\right]}_{q}$ and hyperplane multiplicities $s=r\cdot {\left[k-1\right]}_{q}$, $t=r\cdot {\left[k-1\right]}_{q}+q^{k-1}$, where $r=\sum_{i=1}^{r}\left|S_{i}\right|$.*


Cf. example SU2 in [5]. Field changes work similarly to those explained in Section 6 [5] for two-character sets.

Based on hyperplanes we can construct large families of two-character multisets:

**Lemma 8.** 
*Let $\emptyset \neq H^{\prime }⊊H$ be a subset of the hyperplanes of PG(k−1,q), where k≥3, then*

(3)
$$M=\sum_{H\in H^{\prime }}\chi_{H}$$

*is a two-character multiset with $n=r{\left[k-1\right]}_{q}$, $s=r{\left[k-2\right]}_{q}+q^{k-2}$, and $t=r{\left[k-2\right]}_{q}$, where $r=\#H^{\prime }$.*


By allowing $H^{\prime }$ to be a multiset of hyperplanes we end up with (τ+1)-character sets, where τ is the maximum number of occurrences of a hyperplane in $H^{\prime }$.

Applying Lemma 6 yields:

**Lemma 9.** 
*Let $\emptyset \neq H^{\prime }⊊H$ be a subset of the hyperplanes of PG(k−1,q), where k≥3. If each point P∈P is contained in at least μ∈N elements of $H^{\prime }$, then*

(4)
$$M=\sum_{H\in H^{\prime }}\chi_{H}\;-\;\mu \cdot \chi_{V}$$

*is a two-character multiset with $n=r{\left[k-1\right]}_{q}-\mu {\left[k\right]}_{q}$, $s=r{\left[k-2\right]}_{q}+q^{k-2}-\mu {\left[k-1\right]}_{q}$ and $t=r{\left[k-2\right]}_{q}-\mu {\left[k-1\right]}_{q}$, where $r=|H^{\prime }|$.*


In some cases we obtain two-character multisets where all point multiplicities have a common factor g>1. Here, we can apply the following general construction:

**Lemma 10.** 
*Let M be a multiset of points in PG(k−1,q) such that M(H)∈{s,t} for every hyperplane H∈H. If M(P)≡0(modg) for every point P∈P, then $M^{\prime }:=\frac{1}{g}\cdot M$ is a multiset of points in PG(k−1,q) such that $M^{\prime }\left(H\right)\in \left\{\frac{1}{g}\cdot s,\frac{1}{g}\cdot t\right\}$ for every hyperplane H∈H. Moreover, we have $\#M^{\prime }=\frac{1}{g}\cdot \#M$, $\mu \left(M^{\prime }\right)=\frac{1}{g}\cdot \mu \left(M\right)$, and $\gamma \left(M^{\prime }\right)=\frac{1}{g}\cdot \gamma \left(M\right)$.*


Interestingly enough, it will turn out that we can construct all two-character multisets by combining Lemma 8 with Lemmas 5 and 10; see Theorems 1 and 2.

## 5. Geometric Duals and Sets of Feasible Parameters for Two-Character Multisets

To each two-character multiset M in PG(k−1,q), i.e., $\left\{M(H)\;:\;H\in H\right\}=\left\{s,t\right\}$ for some s,t∈N, we can assign a set of points $\bar{M}$ by using the geometric dual, i.e., interchanging hyperplanes and points. More precisely, fix a non-degenerated billinear form ⊥ and consider pairs of points and hyperplanes (P,H) that are perpendicular with respect to ⊥ (different choices of ⊥ lead to isomorphic configurations). We write $H=P^{⊥}$ for the geometric dual of a point. We define $\bar{M}$ via $\bar{M}\left(P\right)=1$ if M(H)=s, where $H=P^{⊥}$, and $\bar{M}\left(P\right)=0$ otherwise, i.e., if M(H)=t (a generalization of the notion of the geometric dual has been introduced by Brouwer and van Eupen [12] for linear codes and formulated for multisets of points by Dodunekov and Simonis [22]). Of course we have some freedom in how we order *s* and *t*. So, we may also write $\bar{M}\left(P\right)=\left(M(H)-t\right)/\left(s-t\right)\in \left\{0,1\right\}$ for all P∈P, where $H=P^{⊥}$. Noting the asymmetry in *s* and *t* we may also interchange the role of *s* and *t* or replace $\bar{M}$ by its complement. Note that in principle several multisets of points with two hyperplane multiplicities can have the same corresponding point set $\bar{M}$.

For the other direction we can start with an arbitrary set of points $\bar{M}$, i.e., $\bar{M}\left(P\right)\in \left\{0,1\right\}$ for all P∈P. The multiset of points with two hyperplane multiplicities M is then defined via M(H)=s if $\bar{M}\left(P\right)=1$, where $H=P^{⊥}$, and M(H)=t if $\bar{M}\left(P\right)=0$. i.e., we may set
(5)$$M\left(H\right)=t+\left(s-t\right)\cdot \bar{M}\left(H^{⊥}\right).$$
While we have M(H)∈N for all s,t∈N, the point multiplicities M(P) induced by the hyperplane multiplicities M(H) (see Lemma 1) are not integral or non-negative in general. For suitable choices of *s* and *t* they are, while for others they are not.

**Definition 2.** 
*Let $\bar{M}$ be a set of points in PG(k−1,q). By $L\left(\bar{M}\right)\subseteq N^{2}$ we denote the set of all pairs $\left(s,t\right)\in N^{2}$ with s≥t such that a multiset of points M in PG(k−1,q) exists with M(H)=s if $\bar{M}\left(H^{⊥}\right)=1$ and M(H)=t if $\bar{M}\left(H^{⊥}\right)=0$ for all hyperplanes H∈H.*


Directly from Lemma 5 we can conclude:

**Lemma 11.** 
*Let $\bar{M}$ be a set of points in PG(k−1,q). If $\left(s,t\right)\in L\left(\bar{M}\right)$, then we have*

(6)
$${\langle \left(s,t\right)\rangle }_{N}+{\langle \left({\left[k-1\right]}_{q},{\left[k-1\right]}_{q}\right)\rangle }_{N}=\left\{\left(us+v{\left[k-1\right]}_{q},ut+v{\left[k-1\right]}_{q}\right)\;:\;u,v\in N\right\}\subseteq L\left(\bar{M}\right).$$



Before we study the general structure of $L(\bar{M})$ and show that it can be generated by a single element $\left(s_{0},t_{0}\right)$ in the above sense, we consider all non-isomorphic examples in PG(3−1,2) (ignoring the constraint s≥t).

**Example 1.** 
*Let M be a multiset of points in PG(2,2) uniquely characterized by M(L)=s∈N for some line L and $M(L^{\prime })=t\in N$ for all other lines $L^{\prime }\neq L$. For each point P∈L, we have*

(7)
$$M\left(P\right)=\frac{s+2t}{3}-\frac{4t}{6}=\frac{s}{3}$$

*and for each point Q∉L, we have*

(8)
$$M\left(Q\right)=\frac{3t}{3}-\frac{s+3t}{6}=\frac{3t-s}{6}.$$

*Since M(P),M(Q)∈N we set $x:=M\left(P\right)=\frac{s}{3}$ and $y:=M\left(Q\right)=\frac{3t-s}{6}$, so that s=3x and t=2y+x. With this we have n=3x+4y, γ=max{x,y}, and s−t=2(x−y). If x≥y, then we can write $M=y\cdot \chi_{E}+\left(x-y\right)\cdot \chi_{L}$. If x≤y, then we can write $M=y\cdot \chi_{E}-\left(y-x\right)\cdot \chi_{L}$.*


For Example 1 the set of all feasible (s,t)-pairs, assuming s≥t, is given by ${\langle (3,1)\rangle }_{N}+{\langle (3,3)\rangle }_{N}$. If we assume t≥s, then the set of feasible (s,t)-pairs is given by ${\langle (0,2)\rangle }_{N}+{\langle (3,3)\rangle }_{N}$. The vector (0,2) can be computed from (3,1) by computing a suitable complement.

Due to Lemma 6 we can always assume the existence of a point of multiplicity 0 as a normalization. So, in Example 1 we may assume x=0 or y=0, so that $M=y\cdot \chi_{E}-y\cdot \chi_{L}$ or $M=x\cdot \chi_{L}$.

Due to Lemma 10 we can always assume that the greatest common divisor of all point multiplicities is 1 as a normalization (excluding the degenerated case of an empty multiset of points). Applying both normalizations to the multisets of points in Example 1 leaves the two possibilities $\chi_{L}$ and $\chi_{E}-\chi_{L}$, i.e., point sets.

Due to Lemma 7 we always can assume $\#M\leq \gamma \left(M\right)\cdot {\left[k\right]}_{q}/2$. Applying also the third normalization to the multisets of points in Example 1 leaves only the possibility $\chi_{L}$, i.e., a subspace construction; see Proposition 1, where s=3, t=1, n=3, and s−t=2.

**Example 2.** 
*Let M be a multiset of points in PG(2,2) uniquely characterized by $M\left(L_{1}\right)=M\left(L_{2}\right)=s\in N$ for two different lines $L_{1},L_{2}$ and $M(L^{\prime })=t\in N$ for all other lines $L^{\prime }\notin \left\{L_{1},L_{2}\right\}$. For $P:=L_{1}\cap L_{2}$, we have*

(9)
$$M\left(P\right)=\frac{2s+t}{3}-\frac{4t}{6}=\frac{2s-t}{3},$$

*for each point $Q\in \left(L_{1}\cup L_{2}\right)\backslash \left\{P\right\}$, we have*

(10)
$$M\left(Q\right)=\frac{s+2t}{3}-\frac{s+3t}{6}=\frac{s+t}{6},$$

*and for each point $R\notin L_{1}\cup L_{2}$, we have*

(11)
$$M\left(R\right)=\frac{3t}{3}-\frac{2s+2t}{6}=\frac{2t-s}{3}.$$

*Since M(Q),M(R)∈N we set $x:=M\left(Q\right)=\frac{s+t}{6}$ and $y:=M\left(R\right)=\frac{2t-s}{3}$, so that s=4x−y and t=2x+y. With this we have n=6x+7y and s−t=2(x−y). Of course we need to have y≤2x so that M(P)≥0, which implies s≥0.*


*M(P)=0: y=2x, so that M(P)=0, M(Q)=x, M(R)=2x, and the greatest common divisor of M(P), M(Q), and M(R) is equal to x. Thus, we can assume x=1, y=2, so that s=2, t=4, n=8, γ=2, t−s=2, and $M=2\chi_{E}-\chi_{L_{1}}-\chi_{L_{2}}$ for two different lines $L_{1},L_{2}$.*

*M(Q)=0: x=0, so that also y=0 and M is the empty multiset of points.*

*M(R)=0: y=0, M(P)=2x, M(Q)=x, so that gcd(M(P),M(Q),M(R))=x and we can assume x=1. With this we have s=4, t=2, n=6, γ=2, s−t=2, and $M=\chi_{L_{1}}+\chi_{L_{2}}$ for two different lines $L_{1},L_{2}$.*



So, Example 2 can be explained by the construction in Proposition 2.

**Example 3.** 
*Let M be a multiset of points in PG(2,2) uniquely characterized by $M\left(L_{1}\right)=M\left(L_{2}\right)=M\left(L_{3}\right)=s\in N$ for three different lines $L_{1},L_{2},L_{3}$ with a common intersection point $P=L_{1}\cap L_{2}\cap L_{3}$ and $M(L^{\prime })=t\in N$ for all other lines. We have*

(12)
$$M\left(P\right)=\frac{3s}{3}-\frac{4t}{6}=s-\frac{2t}{3}$$

*and*

(13)
$$M\left(Q\right)=\frac{s+2t}{3}-\frac{2s+2t}{6}=\frac{t}{3}$$

*for all points Q≠P. Since M(P),M(Q)∈N we set $x:=M\left(P\right)=s-\frac{2t}{3}$ and $y:=M\left(Q\right)=\frac{t}{3}$, so that s=x+2y and t=3y. With this we have n=x+6y and s−t=x−y.*


*M(P)=0: x=0, so that we can assume y=1, which implies s=2, t=3, γ=1, n=6, t−s=1, and $M=\chi_{E}-\chi_{P}$ for some point P.*

*M(Q)=0: y=0, so that we can assume x=1, which implies s=1, t=0, γ=1, n=1, s−t=1, and $M=\chi_{P}$ for some point P.*



So, also Example 3 can be explained by the subspace construction in Proposition 1.

**Example 4.** 
*Let M be a multiset of points in PG(2,2) uniquely characterized by $M\left(L_{1}\right)=M\left(L_{2}\right)=M\left(L_{3}\right)=s\in N$ for three different lines $L_{1},L_{2},L_{3}$ without a common intersection point, i.e., $L_{1}\cap L_{2}\cap L_{3}=\emptyset$, and $M(L^{\prime })=t\in N$ for all other lines. For each point P that is contained on exactly two lines $L_{i}$, we have*

(14)
$$M\left(P\right)=\frac{2s+t}{3}-\frac{s+3t}{6}=\frac{3s-t}{6},$$

*for each point Q that is contained on exactly one line $L_{i}$, we have*

(15)
$$M\left(Q\right)=\frac{s+2t}{3}-\frac{2s+2t}{6}=\frac{t}{3},$$

*and for the unique point R that is contained on none of the lines $L_{i}$, we have*

(16)
$$M\left(R\right)=\frac{3t}{3}-\frac{3s+t}{6}=\frac{5t-3s}{6}.$$

*Since M(P),M(Q)∈N, we set $x:=M\left(P\right)=\frac{3s-t}{6}$ and $y:=M\left(Q\right)=\frac{t}{3}$, so that s=2x+y and t=3y. With this we have n=2x+5y and s−t=2(x−y).*


*M(P)=0: x=0, so that we can assume y=1, which implies s=1, t=3, t−s=2, γ=2, n=5, and $M=\chi_{L}+2\chi_{P}$ for some line L and some point P∉L.*

*M(Q)=0: y=0, so that x=0 and M is the empty multiset of points.*

*M(R)=0: x=2y, so that we can assume y=1, which implies x=2, s=4, t=6, t−s=2, γ=2, n=9, and the 2-complement of M equals $M=\chi_{L}+2\chi_{P}$ for some line L and some point P∉L; see the case M(P)=0.*



So, also Example 4 can be explained by the construction in Proposition 2.

In Examples 1–4 we have considered all cases of $1\leq \#\bar{M}\leq 3$ up to symmetry. The cases $\#\bar{M}\in \left\{0,7\right\}$ give one-character multisets. By considering the complement $M^{\prime }=\chi_{V}-\bar{M}$ we see that examples for $4\leq \#\bar{M}\leq 6$ do not give anything new. Since the dimension of the ambient space is odd, we cannot apply the construction in Proposition 3.

Now, let us consider the general case. Given the set $\bar{M}$ of hyperplanes with multiplicity *s* we obtain an explicit expression for the multiplicity M(P) of every point P∈P depending on the two unknown hyperplane multiplicities *s* and *t*.

**Lemma 12.** 
*Let $\bar{M}$ be a set of points in PG(k−1,q), where k≥3, and M be a multiset of points in PG(k−1,q) such that M(H)=s if $\bar{M}\left(H^{⊥}\right)=1$ and M(H)=t if $\bar{M}\left(H^{⊥}\right)=0$ for all hyperplanes H∈H. Denoting the number of hyperplanes H∋P with M(H)=s by φ(P) and setting $r:=\#\bar{M}$, Δ:=s−t∈Z, we have*

(17)
$$M\left(P\right)=\frac{t+\Delta \cdot \varphi (P)}{{\left[k-1\right]}_{q}}-\frac{\Delta }{q^{k-2}}\cdot \frac{{\left[k-2\right]}_{q}}{{\left[k-1\right]}_{q}}\cdot \left(r-\varphi (P)\right).$$



**Proof.** Counting gives that ${\left[k-1\right]}_{q}-\varphi \left(P\right)$ hyperplanes through *P* have multiplicity *t*, from the $q^{k-1}$ hyperplanes not containing *P* exactly r−φ(P) have multiplicity M(H)=s and $q^{k-1}-r+\varphi \left(P\right)$ have multiplicity M(H)=t. With this we can use Lemma 1 to compute
$$\begin{array}{ccc} M(P) & = & \sum_{H\in H\;:\;P\in H}\frac{1}{{\left[k-1\right]}_{q}}\cdot M\left(H\right)+\sum_{H\in H\;:\;P\notin H}\frac{1}{q^{k-1}}\cdot \left(\frac{1}{{\left[k-1\right]}_{q}}-1\right)\cdot M\left(H\right)\\ & = & \sum_{H\in H\;:\;P\in H}\frac{1}{{\left[k-1\right]}_{q}}\cdot M\left(H\right)-\sum_{H\in H\;:\;P\notin H}\frac{1}{q^{k-1}}\cdot \frac{q{\left[k-2\right]}_{q}}{{\left[k-1\right]}_{q}}\cdot M\left(H\right)\\ & = & t+\frac{\Delta }{{\left[k-1\right]}_{q}}\cdot \varphi \left(P\right)\;\;-\;\;\;\frac{q{\left[k-2\right]}_{q}}{{\left[k-1\right]}_{q}}\cdot t-\frac{\Delta }{q^{k-1}}\cdot \frac{q{\left[k-2\right]}_{q}}{{\left[k-1\right]}_{q}}\cdot \left(r-\varphi (P)\right)\\ & = & \frac{t+\Delta \cdot \varphi (P)}{{\left[k-1\right]}_{q}}-\frac{\Delta }{q^{k-2}}\cdot \frac{{\left[k-2\right]}_{q}}{{\left[k-1\right]}_{q}}\cdot \left(r-\varphi (P)\right). \end{array}$$□

Note that $\varphi \left(P\right)=\bar{M}\left(P^{⊥}\right)$ for all P∈P.

**Lemma 13.** 
*Let $\bar{M}$ be a set of points in PG(k−1,q), where k≥3, and M be a multiset of points in PG(k−1,q) such that M(H)=s if $\bar{M}\left(H^{⊥}\right)=1$ and M(H)=t if $\bar{M}\left(H^{⊥}\right)=0$ for all hyperplanes H∈H. Denote the number of hyperplanes H∋P with M(H)=s by φ(P) and uniquely choose m∈N, I⊆N with 0∈I such that $\left\{\varphi (P)\;:\;P\in P\right\}=\left\{m+i\;:\;i\in I\right\}$. If s>t and there exists a point Q∈P with M(Q)=0, then we have*

(18)
$$t=\frac{\Delta }{q^{k-2}}\cdot {\left[k-2\right]}_{q}\cdot \left(r-m\right)-\Delta \cdot m$$

*and*

(19)
$$M\left(P\right)=\frac{\Delta \cdot i}{q^{k-2}}$$

*for all points P∈P where i:=φ(P)−m, $r:=\#\bar{M}$, and $\Delta :=s-t\in N_{\geq 1}$. If M is non-repetitive, then *Δ* divides $q^{k-2}$.*


**Proof.** Using Δ>0 we observe that the expression for M(P) in Equation (Equation 17) is increasing in φ(P). So, we need to choose a point Q∈P which minimizes φ(Q) to normalize using M(Q)=0, since otherwise we will obtain points with negative multiplicity. So, choosing a point Q∈P with φ(Q)=m we require
$$0=M\left(Q\right)=\frac{t+\Delta \cdot m}{{\left[k-1\right]}_{q}}-\frac{\Delta }{q^{k-1}}\cdot \frac{q{\left[k-2\right]}_{q}}{{\left[k-1\right]}_{q}}\cdot \left(r-m\right),$$
which yields Equation (Equation 18). Using i:=φ(P)−m and the expression for *t* we compute
$$\begin{array}{ccc} M(P) & = & \frac{t+\Delta \cdot (m+i)}{{\left[k-1\right]}_{q}}-\frac{\Delta }{q^{k-2}}\cdot \frac{{\left[k-2\right]}_{q}}{{\left[k-1\right]}_{q}}\cdot \left(r-m-i\right)\\ & = & \frac{\Delta }{q^{k-2}}\cdot \frac{{\left[k-2\right]}_{q}}{{\left[k-1\right]}_{q}}\cdot \left(r-m\right)-\frac{\Delta \cdot m}{{\left[k-1\right]}_{q}}+\frac{\Delta \cdot (m+i)}{{\left[k-1\right]}_{q}}-\frac{\Delta }{q^{k-2}}\cdot \frac{{\left[k-2\right]}_{q}}{{\left[k-1\right]}_{q}}\cdot \left(r-m-i\right)\\ & = & \frac{\Delta \cdot i}{{\left[k-1\right]}_{q}}+\frac{\Delta \cdot i}{q^{k-2}}\cdot \frac{{\left[k-2\right]}_{q}}{{\left[k-1\right]}_{q}}=\frac{\Delta \cdot i}{q^{k-2}} \end{array}$$
for all P∈P. Note that if f>1 is a divisor of Δ that is coprime to *q*, then all point multiplicities of M are divisible by *f*. If $\Delta =q^{k-2}\cdot f$ for an integer f>1, then all point multiplicities of M are divisible by *f*. Thus, we have that Δ divides $q^{k-2}$. □

Note that $I=\left\{\bar{M}\left(H\right)-\bar{M}\left(H^{\prime }\right)\;:\;H\in H\right\}$, where $H^{\prime }\in H$ is a minimizer of $\bar{M}\left(H\right)$.

**Lemma 14.** 
*Let $\bar{M}$ be a set of points in PG(k−1,q), where k≥3 and M be a multiset of points in PG(k−1,q) such that M(H)=s if $\bar{M}\left(H^{⊥}\right)=1$ and M(H)=t if $\bar{M}\left(H^{⊥}\right)=0$ for all hyperplanes H∈H. Using the notation from Lemma 13 we set*

(20)
$$\begin{array}{ccc} g & = & gcd\;\left(\left\{i\in I\right\}\cup \left\{q^{k-2}\right\}\right), \end{array}$$


(21)
$$\begin{array}{ccc} \Delta_{0} & = & q^{k-2}/g, \end{array}$$


(22)
$$\begin{array}{ccc} t_{0} & = & \frac{1}{g}\cdot {\left[k-2\right]}_{q}\cdot \left(r-m\right)-\Delta_{0}\cdot m,\;\mathrm{and} \end{array}$$


(23)
$$\begin{array}{ccc} s_{0} & = & t+\Delta_{0}. \end{array}$$

*If s>t, then we have*

$$L\left(\bar{M}\right)={\langle (s_{0},t_{0})\rangle }_{N}+{\langle ({\left[k-1\right]}_{q},{\left[k-1\right]}_{q})\rangle }_{N}.$$



**Proof.** Setting μ=μ(M)∈N we have that $M^{\prime }:=M-\mu \cdot \chi_{V}$ is a two-character multiset corresponding to $\left(s^{\prime },t^{\prime }\right):=\left(s-\mu {\left[k-1\right]}_{q},t-\mu {\left[k-1\right]}_{q}\right)\in L\left(\bar{M}\right)$ and there exists a point Q∈P with $M^{\prime }\left(Q\right)=0$. Clearly, we have $\left(s^{\prime },t^{\prime }\right)\in N^{2}$ and $s^{\prime }>t^{\prime }$. From Lemma 13 we conclude the existence of an integer $\Delta^{\prime }\in N_{\geq 1}$ such that $t^{\prime }=\frac{\Delta^{\prime }}{q^{k-2}}\cdot {\left[k-2\right]}_{q}\cdot \left(r-m\right)-\Delta^{\prime }\cdot m$, $s^{\prime }=t^{\prime }+\Delta^{\prime }$, and $M^{\prime }\left(P\right)=\frac{\Delta^{\prime }\cdot i}{q^{k-2}}$ for all P∈P. Since $M^{\prime }\left(P\right)\in N$ for all P∈P we have that $q^{k-2}$ divides $\Delta^{\prime }\cdot g$, so that $\Delta_{0}\in N$ divides $\Delta^{\prime }$. For $f:=\Delta^{\prime }/\Delta_{0}\in N_{\geq 1}$ we observe that $M^{\prime }\left(P\right)$ is divisible by *f* and we set $M^{\prime \prime }:=\frac{1}{f}\cdot M^{\prime }$. With this, we can check that $M^{\prime \prime }$ is a two-character multiset corresponding to $\left(s_{0},t_{0}\right)\in L\left(\bar{M}\right)$. □

Note that it is not necessary to explicitly check $t_{0}\in N$ since $M^{\prime \prime }\left(P\right)\in N$ is sufficient to this end.

Before we consider the problem whether $L\left(\bar{M}\right)\subseteq N^{2}$ contains an element (s,t) with s>t, we treat the so-far-excluded case k=2 separately.

**Lemma 15.** 
*Let $\bar{M}$ be a set of points in PG(1,q). Then, we have*

$$L\left(\bar{M}\right)={\langle (s_{0},0)\rangle }_{N}+{\langle (q+1,q+1)\rangle }_{N},$$

*where $s_{0}=0$ if $\#\bar{M}\in \left\{0,q+1\right\}$ and $s_{0}=1$ otherwise.*


**Proof.** If $\#\bar{M}\in \left\{0,q+1\right\}$, then a two-character multiset M corresponding to $\left(s,t\right)\in \bar{M}$ actually is a one-character multiset and there exists some integer x∈N such that $M=x\cdot \chi_{v}$.Otherwise, we observe that in PG(1,q) points and hyperplanes coincide and the image of $\bar{M}$ is {0,1}. Note that we have $M=t\cdot \chi_{V}+\sum_{P\in P}\left(s-t\right)\cdot \bar{M}\left(P\right)\cdot \chi_{P}$ for each two-character multiset M corresponding to $\left(s,t\right)\in L\left(\bar{M}\right)$ by definition. We can easily check $\left(s,t\right)\in {\langle (1,0)\rangle }_{N}+{\langle (q+1,q+1)\rangle }_{N}$. The proof is completed by choosing s=1 and t=0 in our representation of M. □

**Theorem 1.** 
*Let $\bar{M}$ be a set of points in PG(k−1,q) with $\#\bar{M}\notin \left\{0,{\left[k\right]}_{q}\right\}$, where k≥2. Then,*

(24)
$$M:=\sum_{H\in H}\bar{M}\left(H^{⊥}\right)\cdot \chi_{H}$$

*is a two-character multiset corresponding to $\left(s,t\right)\in L\left(\bar{M}\right)$ with $n=\#M=r{\left[k-1\right]}_{q}$, where $r:=\#\bar{M}$, $t=r{\left[k-2\right]}_{q}$, and $s=r{\left[k-2\right]}_{q}+q^{k-2}$. Setting μ:=μ(M) and $g:=\gcd(\left\{M(P)-\mu \;:\;P\in P\right\})$ the function*

(25)
$$M^{\prime }:=\frac{1}{g}\cdot \left(-\mu \cdot \chi_{V}+\sum_{H\in H}\bar{M}\left(H^{⊥}\right)\cdot \chi_{H}\right)=\frac{1}{g}\cdot \left(M-\mu \cdot \chi_{V}\right)$$

*is a two-character multiset corresponding to $\left(s_{0},t_{0}\right)\in L\left(\bar{M}\right)$ with $n^{\prime }=\#M^{\prime }=\frac{1}{q}\cdot \left(r{\left[k-1\right]}_{q}-\mu {\left[k\right]}_{q}\right)$, where $r:=\#\bar{M}$, $t_{0}=\frac{1}{g}\cdot \left(r{\left[k-2\right]}_{q}-\mu {\left[k-1\right]}_{q}\right)$, and $s_{0}=\frac{1}{g}\cdot \left(r{\left[k-2\right]}_{q}-\mu {\left[k-1\right]}_{q}+q^{k-2}\right)$, and g divides $q^{k-2}$. Moreover, we have*

(26)
$$L\left(\bar{M}\right)={\langle (s_{0},t_{0})\rangle }_{N}+{\langle ({\left[k-1\right]}_{q},{\left[k-1\right]}_{q})\rangle }_{N}\;.$$



**Proof.** We can easily check $M\left(H\right)=r{\left[k-2\right]}_{q}=t$ if $M(H^{⊥})=0$ and $M\left(H\right)=r{\left[k-2\right]}_{q}+q^{k-2}=s$ if $M(H^{⊥})=1$ for all H∈H as well as $\#M=r{\left[k-1\right]}_{q}$ directly from the definition of M. Using Lemmas 6 and 10 we conclude that $M^{\prime }$ is a two-character multiset with the stated parameters.For k=2, Lemma 15 is our last statement. For k≥3 we can apply Lemma 13 to conclude g=gcd({i∈I}) and use the proof of Lemma 14 to conclude our last statement. Since s,t∈N and s>t we have that *g* divides $g\left(s-t\right)=q^{k-2}$. □

Using the notation from Lemma 13 applied to the multiset of points $M-\mu \cdot \chi_{V}$ from Theorem 1 we observe #I≥2 for $\#\bar{M}\notin \left\{0,{\left[k\right]}_{q}\right\}$. Using the facts that $g:=\gcd(\left\{M(P)-\mu \;:\;P\in P\right\})$, that *g* divides $q^{k-2}$, and Equation (Equation 19) we conclude
(27)$$g=gcd\;\left(\{i\in I\}\right)=gcd\;\left(\left\{\bar{M}\left(H\right)-\bar{M}\left(H^{\prime }\right)\;:\;H\in H\right\}\right),$$
where $H^{\prime }\in H$ is a minimizer of $\bar{M}\left(H\right)$.

Using the classification of one-character multisets we conclude from Theorem 1:

**Corollary 1.** 
*Let $\bar{M}$ be a set of points in PG(k−1,q), where k≥2. Then, there exist $\left(s_{0},t_{0}\right)\in N^{2}$ such that $L\left(\bar{M}\right)={\langle (s_{0},t_{0})\rangle }_{N}+{\langle ({\left[k-1\right]}_{q},{\left[k-1\right]}_{q})\rangle }_{N}$.*


**Theorem 2.** 
*Let $\tilde{M}$ be a two-character multiset in PG(k−1,q), where k≥2. Then, there exist unique u,v∈N such that $\tilde{M}=u\cdot M^{\prime }+v\cdot \chi_{V}$, where $M^{\prime }$ is given by Equation (Equation 25).*


**Proof.** Let s>t be the two hyperplane multiplicities of $\tilde{M}$. With this, define $\bar{M}$ such that $\bar{M}\left(H^{⊥}\right)=1$ if $\tilde{M}\left(H\right)=s$ and $\bar{M}\left(H^{⊥}\right)=0$ if $\tilde{M}\left(H\right)=t$ for all H∈H. So, $\left(s,t\right)\in L\left(\bar{M}\right)$ and Theorem 1 yields the existence of u,v∈N with $\left(s,t\right)=u\cdot \left(s_{0},t_{0}\right)+v\cdot \left({\left[k-1\right]}_{q},{\left[k-1\right]}_{q}\right)$, where $s_{0}$, $t_{0}$ are as in Theorem 1. From Lemma 1 we then conclude $\tilde{M}=u\cdot M^{\prime }+v\cdot \chi_{V}$. Note that $\mu (M^{\prime })$ and $\mu (\chi_{V})=1$ imply $\mu (\tilde{M})=v$, so that *u* can be computed from $\gamma \left(\tilde{M}\right)=u\cdot \gamma \left(M^{\prime }\right)+v$, i.e., *u* and *v* are uniquely determined. □

Note that for a one-character multiset $\tilde{M}$ there exists a unique v∈N such that $\tilde{M}=v\cdot \chi_{V}$. Given a set of points $\bar{M}$ we call $M^{\prime }$ the canonical representant of the set of two-character multisets M corresponding to $\left(s,t\right)\in L\left(\bar{M}\right)$. If $M=M^{\prime }$ we just say that M is the canonical two-character multiset.

**Theorem 3.** 
*Let $w_{1}<w_{2}$ be the non-zero weights of a non-repetitive ${\left[n,k\right]}_{q}$ two-weight code C without full support. Then, there exist integers f and u such that $w_{1}=up^{f}$ and $w_{2}=\left(u+1\right)p^{f}$, where p is the characteristic of the underlying field $F_{q}$.*


**Proof.** Let M be the two-character multiset in PG(k−1,q) corresponding to *C*. Choose unique u,v∈N such that $M=u\cdot M^{\prime }+v\cdot \chi_{V}$, as in Theorem 2. Since *C* does not have full support, we have v=0 and since *C* is non-repetitive we have u=1. With this we can use Theorem 1 to compute
(28)$$w_{1}=n-s=\frac{1}{g}\cdot \left(r\cdot q^{k-2}-\mu \cdot q^{k-1}-q^{k-2}\right)=\left(r-q\mu -1\right)\cdot p^{f}$$
and
(29)$$w_{2}=n-t=\frac{1}{g}\cdot \left(r\cdot q^{k-2}-\mu \cdot q^{k-1}\right)=\left(r-q\mu \right)\cdot p^{f},$$
where *f* is chosen such that $\frac{q^{k-2}}{g}=p^{f}$, i.e., we can choose u=r−qμ−1. □

We have seen in Equation (Equation 27) that we can compute the parameter *g* directly from the set of points $\bar{M}$. If we additionally assume that $\bar{M}$ is spanning, then we can consider the corresponding projective ${\left[n,k\right]}_{q}$-code $\bar{C}$, where $n=\#\bar{M}$ (if $\bar{M}$ is not spanning, then we can consider the lower-dimensional subspace spanned by $\mathrm{supp}(\bar{M})$). Note that we have $\bar{M}\left(H\right)\equiv m\;\left(\mathrm{mod}\;g\right)$ for all H∈H and that *g* is maximal with this property. If m≡n(modg), then *g* would simply be the maximal divisibility constant of the weights of $\bar{C}$. From theorem 7 in [24] or theorem 3 in [25] we can conclude m≡n(modg). Thus, we have
(30)$$g=\gcd\;\left(\left\{\mathrm{wt}\left(c\right)\;:\;c\in \bar{C}\right\}\right).$$
The argument may also be based on the following lemma (using the fact that $\bar{C}$ is projective):

**Lemma 16.** 
*Let C be an ${\left[n,k\right]}_{q}$-code of full length such that we have wt(c)≡m(modΔ) for all non-zero codewords c∈C. If *Δ* is a power of the characteristic of the underlying field $F_{q}$, then we have m≡0(modmin{Δ,q}). Moreover, if additionally q divides *Δ* and k≥2, then the non-zero weights in each residual code are congruent to m/q modulo Δ/q.*


**Proof.** Let M be the multiset of points in PG(k−1,q) corresponding to *C*. For each hyperplane *H* we have n−M(H)≡m(modΔ), which is equivalent to M(H)≡n−m(modΔ). The weight of a non-zero codeword in a residual code is given by a subspace *K* of codimension 2 and a hyperplane *H* with K≤H. With this, the weight is given by M(H)−M(K)≡n−m−M(K)(modΔ). Counting the hyperplane multiplicities of the q+1 hyperplanes that contain *K* yields
(31)$$\sum_{H\in H\;:\;K\leq H}M\left(H\right)=\#M+q\cdot M\left(K\right)=\#M+q\cdot M\left(K\right)$$
and
(32)$$\sum_{H\in H\;:\;K\leq H}M\left(H\right)\equiv \left(q+1\right)\left(n-m\right)\;\left(\mathrm{mod}\;\Delta \right),$$
so that
(33)$$m\equiv q\cdot \left(n-m-M(K)\right)\;\left(\mathrm{mod}\;\Delta \right).$$□

Given Equation (Equation 30) we might be interested in projective divisible codes (with a large divisibility constant). For enumerations for the binary case we refer the reader to [26] and for a more general survey we refer the reader to, e.g., [27]. Note that the only point sets M in PG(k−1,q) that are $q^{k-1}$-divisible are given by $\#M\in \left\{0,{\left[k\right]}_{q}\right\}$, i.e., the empty and the full set. All other point sets are at most $q^{k-2}$-divisible, as implied by Theorem 1.

## 6. Enumeration of Two-Character Multisets in PG(k−1,q) for Small Parameters

Since all two-character multisets in PG(1,q) can be parameterized as $M=b\cdot \chi_{V}+\sum_{P\in P}\left(a-b\right)\cdot \bar{M}\left(P\right)\cdot \chi_{P}$ for integers a>b≥0 and a set of points $\bar{M}$ in PG(k−1,q) (see Lemma 15 and its proof), we assume k≥3 in the following. Due to Theorem 2, every two-character multiset in PG(k−1,q) can be written as $u\cdot M^{\prime }+v\cdot \chi_{V}$, where u,v∈N and $M^{\prime }$ is characterized in Theorem 1. So, we further restrict out considerations on canonical two-character multisets where we have u=1 and v=0. For k=2, all canonical two-character multisets in PG(k−1,q) are indeed sets of points and given by the construction in Proposition 3 (with r=1).

It can be easily checked that two isomorphic sets of points in PG(k−1,q) yield isomorphic canonical two-character multisets $M^{\prime }$. So, for the full enumeration of canonical two-character multisets in PG(k−1,q) we just need to loop over all non-isomorphic sets of points $\bar{M}$ in PG(k−1,q) and use Theorem 1 to determine M, $M^{\prime }$, and their parameters. We remark that the numbers of non-isomorphic projective codes per length, dimension, and field size are, e.g., listed in tables 6.10–6.12 in [28] (for small parameters). For the binary case and at most six dimensions some additional data can be found in [29]. Here, we utilize the software package LinCode [30] to enumerate these codes.

In Table 1 and Table 2 we list the feasible parameters for canonical two-character multisets in PG(2,2) and in PG(3,2), respectively, where $n^{\prime }:=\#M^{\prime }$ and $\gamma^{\prime }:=\gamma \left(M^{\prime }\right)$. The two hyperplane multiplicities for $M^{\prime }$ are denoted by $s_{0}$, $t_{0}$ and those of M by s,t. The parameters *g*, μ, *r* are as in (Equation 25) and n=#M. For PG(2,2) we can also state more direct constructions:$\left(n^{\prime },s_{0},t_{0},\gamma^{\prime }\right)=\left(1,1,0,1\right)$: characteristic function of a point (not spanning);$\left(n^{\prime },s_{0},t_{0},\gamma^{\prime }\right)=\left(3,3,1,1\right)$: characteristic function of a line (not spanning);$\left(n^{\prime },s_{0},t_{0},\gamma^{\prime }\right)=\left(4,2,0,1\right)$: complement of the characteristic function of a line;$\left(n^{\prime },s_{0},t_{0},\gamma^{\prime }\right)=\left(6,3,2,1\right)$: complement of the characteristic function of a point;$\left(n^{\prime },s_{0},t_{0},\gamma^{\prime }\right)=\left(5,3,1,2\right)$: $\chi_{L}+2\chi_{P}$ for a line *L* and a point *P* with P∉L;$\left(n^{\prime },s_{0},t_{0},\gamma^{\prime }\right)=\left(6,4,2,2\right)$: $\chi_{L}+\chi_{L}^{\prime }$ for two different lines *L* and $L^{\prime }$;$\left(n^{\prime },s_{0},t_{0},\gamma^{\prime }\right)=\left(8,4,2,2\right)$: $\chi_{V}-\chi_{L}-\chi_{L}^{\prime }$ for two different lines *L* and $L^{\prime }$;$\left(n^{\prime },s_{0},t_{0},\gamma^{\prime }\right)=\left(9,5,3,2\right)$: $2\chi_{V}-\chi_{L}+-\chi_{P}$ for a line *L* and a point *P* with P∉L.

Of course, also for PG(3,2) some of the examples have nicer descriptions:$\left(n^{\prime },s_{0},t_{0},\gamma^{\prime }\right)=\left(1,1,0,1\right)$: characteristic function of a point (not spanning);$\left(n^{\prime },s_{0},t_{0},\gamma^{\prime }\right)=\left(3,3,1,1\right)$: characteristic function of a line (not spanning);$\left(n^{\prime },s_{0},t_{0},\gamma^{\prime }\right)=\left(7,7,3,1\right)$: characteristic function of a plane (not spanning);$\left(n^{\prime },s_{0},t_{0},\gamma^{\prime }\right)=\left(5,3,1,1\right)$: projective base; spanning projective 2-weight code;$\left(n^{\prime },s_{0},t_{0},\gamma^{\prime }\right)=\left(6,4,2,1\right)$: characteristic function of two disjoint lines; spanning projective 2-weight code;$\left(n^{\prime },s_{0},t_{0},\gamma^{\prime }\right)=\left(14,10,6,2\right)$: characteristic function of two different planes;$\left(n^{\prime },s_{0},t_{0},\gamma^{\prime }\right)=\left(21,13,9,3\right)$: characteristic function of three planes intersecting in a common point but not a common line.

Note that we may restrict our considerations to $r<{\left[k\right]}_{q}/2$, since if $M^{\prime }$ is the a canonical two-character multiset for a set of points $\bar{M}$ with $\#\bar{M}=r$, then the complement of $M^{\prime }$ is the a canonical two-character multiset for a set of points which is the complement of $\bar{M}$ and has cardinality ${\left[k\right]}_{q}-r$.

From the data in Table 1 and Table 2 we can guess the maximum possible point multiplicity $\gamma (M^{\prime })$ of $M^{\prime }$:

**Proposition 4.** 
*Let M be a canonical two-character multiset in PG(k−1,q), where k≥2. Then, we have $\gamma \left(M\right)\leq q^{k-2}$.*


**Proof.** Choose a suitable set $H^{\prime }\subseteq H$ and g,ν∈N such that
$$M=\frac{1}{g}\cdot \left(\sum_{H\in H^{\prime }}\chi_{H}\;-\;\mu \cdot \chi_{V}\right).$$
Let P∈P be a point with M(P)=γ and Q∈P be a point with M(Q)=0. With this we have $\lambda \geq \left|\left\{H\in H^{\prime }\;:\;Q\leq H\right\}\right|$. Since *P* is contained in ${\left[k-1\right]}_{q}$ hyperplanes in H and 〈P,Q〉 is contained in ${\left[k-2\right]}_{q}$ hyperplanes in H, we have $M\left(P\right)\leq q^{k-2}$. □

We can easily construct an example showing that the stated upper bound is tight. To this end, let *P*, *Q* be two different points in PG(k−1,q), where k≥3, and $H^{\prime }$ be an arbitrary hyperplane neither containing *P* nor *Q*. With this, we choose $H^{\prime }$ as the set of all $q^{k-2}$ hyperplanes that contain *P* but do not contain *Q* and additionally the hyperplane $H^{\prime }$. For the corresponding multiset of points M we then have $M\left(P\right)=q^{k-2}$ and M(Q)=0, so that μ(M)=0. For an arbitrary point $R\in H^{\prime }$ we have $M\left(R\right)=q^{k-2}-q^{k-3}+1=\left(q-1\right)q^{k-3}+1$, so that gcd(M(R),M(P))=1 if k≥4 or k=3 and q≠2. For (k,q)=(3,2) we have already seen examples of canonical two-character multisets with maximum point multiplicity 2.

In Table 3 and Table 4, we list the feasible parameters for canonical two-character multisets in PG(4,2) with point multiplicity at most 4.

## Figures and Tables

**Table 1 entropy-26-00289-t001:** Feasible parameters for canonical two-character multisets in PG(2,2).

*g*	μ	*r*	*n*	γ	*s*	*t*	$s_{0}$	$t_{0}$	$n^{\prime }$	$\gamma^{\prime }$
2	1	3	9	3	5	3	1	0	1	1
1	0	1	3	1	3	1	3	1	3	1
1	2	6	18	3	8	6	2	0	4	1
2	0	4	12	2	6	4	3	2	6	1
1	1	4	12	3	6	4	3	1	5	2
1	0	2	6	2	4	2	4	2	6	2
1	1	5	15	3	7	5	4	2	8	2
1	0	3	9	2	5	3	5	3	9	2

**Table 2 entropy-26-00289-t002:** Feasible parameters for canonical two-character multisets in PG(3,2).

*g*	μ	*r*	*n*	γ	*s*	*t*	$s_{0}$	$t_{0}$	$n^{\prime }$	$\gamma^{\prime }$
4	3	7	49	7	25	21	1	0	1	1
2	1	3	21	3	13	9	3	1	3	1
2	4	10	70	6	34	30	3	1	5	1
2	2	6	42	4	22	18	4	2	6	1
1	0	1	7	1	7	3	7	3	7	1
1	6	14	98	7	46	42	4	0	8	1
2	3	9	63	5	31	27	5	3	9	1
2	1	5	35	3	19	15	6	4	10	1
2	4	12	84	6	40	36	6	4	12	1
4	0	8	56	4	28	24	7	6	14	1
2	2	8	56	6	28	24	7	5	13	2
1	0	2	14	2	10	6	10	6	14	2
2	0	4	28	4	16	12	8	6	14	2
1	5	13	91	7	43	39	8	4	16	2
2	3	11	77	7	37	33	8	6	16	2
2	1	7	49	5	25	21	9	7	17	2
1	1	4	28	4	16	12	9	5	13	3
1	4	11	77	7	37	33	9	5	17	3
1	3	9	63	6	31	27	10	6	18	3
1	2	7	49	5	25	21	11	7	19	3
1	1	5	35	4	19	15	12	8	20	3
1	0	3	21	3	13	9	13	9	21	3
1	4	12	84	7	40	36	12	8	24	3
1	3	10	70	6	34	30	13	9	25	3
1	2	8	56	5	28	24	14	10	26	3
1	1	6	42	4	22	18	15	11	27	3
1	0	4	28	3	16	12	16	12	28	3
1	3	11	77	6	37	33	16	12	32	3
1	3	8	56	7	28	24	7	3	11	4
1	2	6	42	6	22	18	8	4	12	4
1	3	9	63	7	31	27	10	6	18	4
1	2	7	49	6	25	21	11	7	19	4
1	1	5	35	5	19	15	12	8	20	4
1	3	10	70	7	34	30	13	9	25	4
1	2	8	56	6	28	24	14	10	26	4
1	1	6	42	5	22	18	15	11	27	4
1	2	9	63	6	31	27	17	13	33	4
1	1	7	49	5	25	21	18	14	34	4
1	0	5	35	4	19	15	19	15	35	4
1	2	10	70	6	34	30	20	16	40	4
1	1	8	56	5	28	24	21	17	41	4
1	0	6	42	4	22	18	22	18	42	4
1	1	9	63	5	31	27	24	20	48	4
1	0	7	49	4	25	21	25	21	49	4

**Table 3 entropy-26-00289-t003:** Feasible parameters for canonical two-character multisets in PG(4,2) with $\gamma^{\prime }\leq 4$—part 1.

*g*	μ	*r*	*n*	γ	*s*	*t*	$s_{0}$	$t_{0}$	$n^{\prime }$	$\gamma^{\prime }$
8	7	15	225	15	113	105	1	0	1	1
4	3	7	105	7	57	49	3	1	3	1
2	1	3	45	3	29	21	7	3	7	1
1	0	1	15	1	15	7	15	7	15	1
1	14	30	450	15	218	210	8	0	16	1
2	12	28	420	14	204	196	12	8	24	1
4	8	24	360	12	176	168	14	12	28	1
8	0	16	240	8	120	112	15	14	30	1
2	2	6	90	6	50	42	10	6	14	2
2	5	13	195	9	99	91	12	8	20	2
2	3	9	135	7	71	63	13	9	21	2
2	1	5	75	5	43	35	14	10	22	2
2	8	20	300	12	148	140	14	10	26	2
2	6	16	240	10	120	112	15	11	27	2
2	4	12	180	8	92	84	16	12	28	2
2	2	8	120	6	64	56	17	13	29	2
4	0	8	120	8	64	56	16	14	30	2
2	0	4	60	4	36	28	18	14	30	2
1	0	2	30	2	22	14	22	14	30	2
4	7	23	345	15	169	161	16	14	32	2
2	11	27	405	15	197	189	16	12	32	2
1	13	29	435	15	211	203	16	8	32	2
2	9	23	345	13	169	161	17	13	33	2
2	7	19	285	11	141	133	18	14	34	2
2	5	15	225	9	113	105	19	15	35	2
2	3	11	165	7	85	77	20	16	36	2
2	10	26	390	14	190	182	20	16	40	2
2	8	22	330	12	162	154	21	17	41	2
2	6	18	270	10	134	126	22	18	42	2
2	9	25	375	13	183	175	24	20	48	2
2	4	10	150	10	78	70	9	5	13	3
2	9	21	315	15	155	147	10	6	18	3
2	3	9	135	9	71	63	13	9	21	3
2	6	16	240	12	120	112	15	11	27	3
2	4	12	180	10	92	84	16	12	28	3
2	2	8	120	8	64	56	17	13	29	3
1	1	4	60	4	36	28	21	13	29	3
2	7	19	285	13	141	133	18	14	34	3
2	5	15	225	11	113	105	19	15	35	3
2	3	11	165	9	85	77	20	16	36	3
2	1	7	105	7	57	49	21	17	37	3
2	6	18	270	12	134	126	22	18	42	3
2	4	14	210	10	106	98	23	19	43	3
2	2	10	150	8	78	70	24	20	44	3
1	0	3	45	3	29	21	29	21	45	3
1	12	28	420	15	204	196	24	16	48	3
2	7	21	315	13	155	147	25	21	49	3

**Table 4 entropy-26-00289-t004:** Feasible parameters for canonical two-character multisets in PG(4,2) with $\gamma^{\prime }\leq 4$—part 2.

*g*	μ	*r*	*n*	γ	*s*	*t*	$s_{0}$	$t_{0}$	$n^{\prime }$	$\gamma^{\prime }$
2	5	17	255	11	127	119	26	22	50	3
2	3	13	195	9	99	91	27	23	51	3
2	8	24	360	14	176	168	28	24	56	3
2	6	20	300	12	148	140	29	25	57	3
2	4	16	240	10	120	112	30	26	58	3
2	2	12	180	8	92	84	31	27	59	3
1	11	27	405	14	197	189	32	24	64	3
2	7	23	345	13	169	161	32	28	64	3
2	5	19	285	11	141	133	33	29	65	3
2	3	15	225	9	113	105	34	30	66	3
2	6	22	330	12	162	154	36	32	72	3
2	0	10	150	6	78	70	39	35	75	3
2	5	21	315	11	155	147	40	36	80	3
2	4	12	180	12	92	84	16	12	28	4
1	2	6	90	6	50	42	20	12	28	4
2	3	11	165	11	85	77	20	16	36	4
2	4	14	210	12	106	98	23	19	43	4
1	2	7	105	6	57	49	27	19	43	4
1	1	5	75	5	43	35	28	20	44	4
1	11	26	390	15	190	182	25	17	49	4
1	10	24	360	14	176	168	26	18	50	4
1	9	22	330	13	162	154	27	19	51	4
1	8	20	300	12	148	140	28	20	52	4
1	7	18	270	11	134	126	29	21	53	4
1	5	14	210	9	106	98	31	23	55	4
2	6	20	300	14	148	140	29	25	57	4
1	3	10	150	7	78	70	33	25	57	4
2	4	16	240	12	120	112	30	26	58	4
1	2	8	120	6	64	56	34	26	58	4
1	1	6	90	5	50	42	35	27	59	4
1	0	4	60	4	36	28	36	28	60	4
1	11	27	405	15	197	189	32	24	64	4
1	10	25	375	14	183	175	33	25	65	4
2	3	15	225	11	113	105	34	30	66	4
1	9	23	345	13	169	161	34	26	66	4
2	1	11	165	9	85	77	35	31	67	4
1	8	21	315	12	155	147	35	27	67	4
1	6	17	255	10	127	119	37	29	69	4
1	4	13	195	8	99	91	39	31	71	4
1	3	11	165	7	85	77	40	32	72	4
1	2	9	135	6	71	63	41	33	73	4
1	1	7	105	5	57	49	42	34	74	4
1	0	5	75	4	43	35	43	35	75	4
1	10	26	390	14	190	182	40	32	80	4
2	3	17	255	11	127	119	41	37	81	4
1	9	24	360	13	176	168	41	33	81	4
2	4	20	300	12	148	140	44	40	88	4
2	3	19	285	11	141	133	48	44	96	4
1	9	25	375	13	183	175	48	40	96	4

## Data Availability

No new data were created or analyzed in this study. Data sharing is not applicable to this article.

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
