# Peer review of "Non-Projective Two-Weight Codes"

_entropy, 2024, doi:10.3390/e26040289_

Round 1

Reviewer 1 Report

Comments and Suggestions for Authors

I have seen ths paper. It can be published.

It is a long paper. Part of the length is caused by the many examples.
I would not mind if the paper were shortened by removing some of those.

There are lengthy tables. I did not discover precisely what is in the tables: perhaps one has to read all of the text to discover what the symbols mean. It would be nice to have a compact description of the meaning of the notation ("These tables have 11 columns with header g, mu, r, n, gamma, s, t, s0, t0, n', gamma'. These numbers are ... Following tables have 9 columns with header r, mu, g, n', s0, t0, a_s, a_t, construction. The numbers are ...") so that one could consult the tables without reading all text.

The paper fixes, completely arbitrarily, a non-degenerate bilinear form to create a correspondence between points and hyperplanes. But this correspondence does not play any role, and the same text could be written shorter and more clearly without it.

Author Response

Thanks for your report. The paper is indeed rather long. However since several statements are quite technical I would like to keep the four explicit examples if you do not insist otherwise. 

The parameters of the tables are now explained directly before refering to those tables. Thanks for pointing out. 

The remark on the non-degenerate bilinear form has been clarified now.

Reviewer 2 Report

Comments and Suggestions for Authors

\textbf{\large{Review Report for  ``Non-projective two-weight codes"}}

In this work, the authors studied non-projective  two-weight codes and showed the result under mild extra conditions. For small dimensions they gave exhaustive enumerations of the feasible parameters in the binary case.

Some comments are given as follows.\\

The results presented in the manuscript make good progress on two-weight codes. Non-projective two-weight codes over $\mathbb{F}_{q}$ are classified by the constructions for two-character multisets of points. I think it can be accepted for publication in Entropy after considering the following suggestions.

\begin{enumerate}

\item \textbf{Page 4}.  In the proof of \textbf{Lemma 3.2}, please  provide detailed proof of this equation

$$\sum_{H\in \mathcal{H}:P\in H}\chi_{H}-\frac{[k-2]_q}{[k-1]_q}\cdot\sum_{H\in \mathcal{H}}\chi_{H}=q^{k-2}\chi_{P}.$$

\item  \textbf{Pages 5-6}. Several Lemmas and Propositions are given here. Although some of them are easily obtained, I advise to give either simple  proofs or  cited references.

\item \textbf{Page 11}. Please provide an explanation or proof for the uniqueness of $u,v\in\mathbb{N}$.

\item There exist typos and syntax errors in the manuscript. Please examine it carefully. I list a part of them here. In Line 2 in Abstract, ``two-weight" should be ``two-weight code"; in Line 226, ``can generated" should be ``can be generated"; in Line 334, ``exist" should be ``exists"; in Line 345, it should use a period at the end of the equation; in Line 438, delete a redundant ``the".

\end{enumerate}

Comments on the Quality of English Language

The paper is well written 

Author Response

Thanks a lot for your report. 

In the proof of Lemma 3.2 more details are given for the stated equation. 

Yes, you are right that some lemmas and propositions on pages 5-6 don't have explicit proofs. Since the other referee criticized the length of the paper, I would prefer not to given explicit proofs for these more or less obvious statements.

The uniqueness of u and v is now explained better.

Thanks for your explicit list of typos. These and several further ones are corrected now.